# Comparison of conventional versus customised Eurosil-4 Pink bolus for radiotherapy of the chest wall

**Ashlesha Gill**[1]*, **Warwick Smith**[2], **Andrew Hirst**[1], **Mahsheed Sabet**[1,2], **Zaid Alkhatib**[2], **Suki Gill**[1,2], **Pejman Rowshanfarzad**[1]

1 School of Physics, Mathematics and Computing, The University of Western Australia, Perth, WA, Australia,
2 Department of Radiation Oncology, Sir Charles Gairdner Hospital (SCGH), Perth, WA, Australia

* gill.ashlesha@gmail.com

**Data Availability Statement:** All relevant data are within the manuscript and its Supporting Information files.

## Abstract

### Introduction

*In radiotherapy, the presence of* air gaps near a tumour can lead to underdose to the tumour. In this study, the impact of air gaps on dose to the surface was evaluated. 3D-printing was used to construct a Eurosil-4 Pink bolus customised to the patient and its dosimetric properties were compared with that of Paraffin wax bolus.

### Methods

Surface dose was measured for flat sheets of Eurosil-4 Pink bolus with different thicknesses. Different air gap thicknesses were inserted between the bolus and the surface, and dose was measured for each air gap using 10 cm × 10 cm fields. This was repeated with the effective field size calculated from the patient plan. Surface dose was measured for varying angles of incidence. A customised chest phantom was used to compare dose for two customised Eurosil-4 Pink boluses, and commonly used Paraffin wax bolus.

### Results

The surface dose was found to be highest for 1.1 cm thick bolus. The decrease in surface dose for the Eurosil-4 Pink bolus was minimal for the 10 cm × 10 cm field, but higher for the effective field size and larger angles of incidence. For instance, the dose was reduced by 6.2% as a result of 1 cm air gap for the effective field size and 60 degree angle of incidence. The doses measured using Gafchromic film under the customised Eurosil-4 Pink boluses were similar to that of the Paraffin wax bolus, and higher than prescribed dose.

### Conclusions

The impact of air gaps can be significant for small field sizes and oblique beams. A customised Eurosil-4 Pink bolus has promising physical and dosimetric properties to ensure sufficient dose to the tumour, even for treatments where larger impact of air gaps is suspected.

**Funding:** The author(s) received no specific funding for this work.

**Competing interests:** The authors have declared that no competing interests exist.

## Introduction

Chest wall is the recurrence site in more than half of the breast cancer patients who experience locoregional failure [1]. Post-mastectomy radiotherapy is recommended for the chest wall using megavoltage photon and electron beams. ICRU Report 50 specifies that the dose delivered to the target volume should be kept within +7% and -5% of the prescribed dose [2]. For the high energy beams used in radiotherapy, the surface dose is reduced significantly due to the skin sparing effect, which necessitates the use of a bolus. Air gaps are observed under the bolus if it cannot conform precisely to the skin surface. It is not possible to account for air gaps in the treatment planning system (TPS), as the location of the air gaps cannot be ascertained before the actual setup [3]. Thus, it is imperative to assess the impact of air gaps on the surface dose and optimise the use of bolus in the clinical setting.

Several studies have demonstrated a decrease in the surface dose associated with the air gaps between the bolus and skin, along with its dependence on the treatment parameters set for the photon beam [4–6]. Butson *et al.* showed that the skin dose was reduced by up to 10% in the presence of 1 cm air gap, for a field size of 8 cm × 8 cm, and an angle of incidence of 60 degrees [4]. Sroka *et al.* analysed the percentage depth dose (PDD) for varying distance between the bolus and the surface [5]. They showed that the depth of maximum dose becomes larger as the bolus is moved up, the effect being larger for small fields. Khan *et al.* found that the surface dose reduced considerably for air gaps larger than 0.5 cm when a 5 cm × 5 cm field was used and emphasized the need for bolus conformity in the case of small fields [6].

Some studies have reported the use of 3D-printing to make a bolus customised to the patient surface [7, 8]. The CT of the patient was imported into the TPS to construct a conformal bolus, which was thereafter modified to a printable format. Kim *et al.* used the method to create a bolus composed of acrylonitrile butadiene styrene (ABS) for the RANDO phantom and found that the bolus provided suitable dose escalation to the surface [7]. Fujimoto *et al.* showed that the 3D-printed ABS bolus was capable of reducing air gaps and ensuring dose coverage [8]. This provides an alternative to the commercial bolus, but further research is possible to establish its usefulness. Paraffin wax bolus is commonly used in the clinic, but since it is completely flat, there is an increased chance of having air gaps when placed on irregular body surfaces. The customised 3D-printed boluses can rectify this problem and improve setup reproducibility [7, 8]. In the present study, it is suggested that a hollow shell specifically made for each patient based on the CT images be 3D-printed and filled with the bolus material.

The study consists of two parts: quantification of the effect of air gaps for a flat Eurosil-4 Pink bolus; and investigation of the clinical feasibility of using customised Eurosil-4 Pink bolus created for each patient. The dose reduction caused by the presence of air gaps was checked for different bolus thicknesses, air gaps, field sizes and angles of beam incidence. Finally, the surface doses using the customised Eurosil-4 Pink bolus and Paraffin wax bolus were compared.

## Materials and methods

### Air gap analysis

For the purpose of finding the optimal bolus thickness, Eurosil-4 Pink bolus sheets (Fig 1(A)) of 0.5–4 cm thickness in 0.5 cm steps were modelled using a 10 cm × 10 cm × 5 cm tray. The same process was repeated to make a single sheet with an insert for a CC04 ionisation chamber (IBA Dosimetry, Germany) to fit it close to the surface. The CT scans of the boluses were used to check their uniformity of thickness and density. Each bolus sheet was placed on the sheet containing the ionisation chamber, and 10 cm of solid water was used for backscatter.

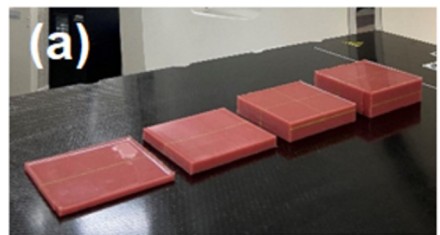
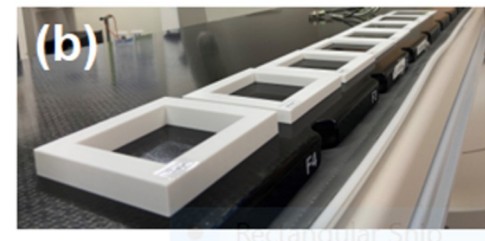
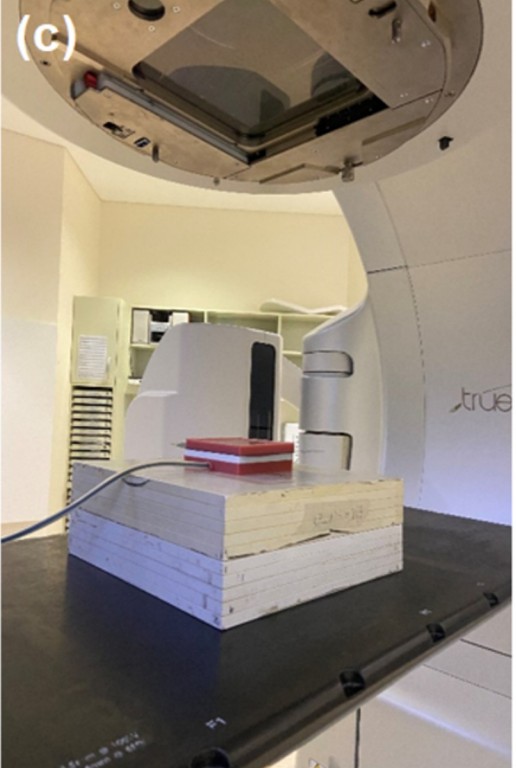
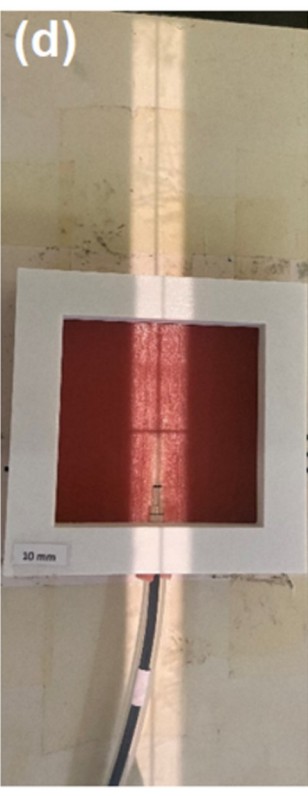

**Fig 1.** (a) Flat Eurosil-4 Pink boluses of different thickness (b) Stands for varying size of air gap (c) Setup for air gap measurements with 1 cm bolus (d) Effective field size from the VMAT patient plan (2.8 cm × 22.7 cm).

Treatment plans were generated using the Eclipse Treatment planning system (ver. 15.6, Varian Medical Systems, Palo Alto, CA), for 200 monitor units (MUs) of 6 MV photon beams. Measurements were made using 10 cm × 10 cm fields, and a 100 cm source to axis distance (SAD). Surface dose was calculated using the Acuros XB algorithm. Plans were delivered on a True Beam linear accelerator (Varian Medical Systems, Palo Alto, CA), and point doses were measured with a CC04 ionisation chamber in conjunction with a PTW Unidos electrometer (PTW, Freiburg, Germany).

Air gap stands of thicknesses: 0.1, 0.2, 0.3, 0.4, 0.5, 0.76, 1.0 and 1.5 cm were 3D printed using a Prusa i3 MK3S 3D-Printer (Prusa Research, Czech Republic) (Fig 1(B)). Air gaps were inserted between the bolus and the sheet containing the ion chamber (Fig 1(C)). Treatment plans were delivered for each air gap thickness and doses were measured with the CC04 ion chamber.

The patient plan was used to find the gap size between each individual leaf pair for 114 control points. The average gap size was found to be 2.8 cm, and the average beam size was 63.6

**Fig 2.** (a) Bolus designed in Meshmixer (b) 3D-printed PLA bolus shell (c) Removal of the shell (d) Resulting customised Eurosil-4 Pink bolus.

$cm^2$. Therefore, it was established that the effective beam size used for the patient was 2.8 cm × 22.7 cm (Fig 1(D)). The previous air gap experiment was repeated with 2.8 cm × 22.7 cm field and dose was measured in the ion chamber. Then, gantry angle was changed to 0, 10, 20, 30, 40, 50 and 60 degrees, and for each angle of incidence, dose was measured with the ion chamber for 1 cm air gap and no air gap. Other treatment parameters were kept unchanged.

## Construction of a customised Eurosil-4 Pink bolus

The CT of the patient was imported into 3D Slicer ver. 4.11 (Brigham and Women's Hospital, Boston, MA, USA) to carry out volume rendering of the chest region and convert it into stereolithography (STL) file format [9]. The Tumour STL model was imported into Mesh Mixer (Autodesk, San Rafael, CA, USA). The surface of this model was selected, extracted, and then extruded, to a predefined thickness of 0.6 cm and 1.1 cm. This layer was then smoothened to remove artefacts and CT slice irregularities. The resultant STL file (Fig 2(A)) provided the required 3D model for construction of a bolus shell that could be used to customise the Euro-sil-4 Pink bolus (SynTec, Schouten Group, Netherlands). This model was transferred to the slicing software, Prusa Slicer (Prusa Research, Czech Republic) and the resulting G-code was sent to the 3D printer. The bolus model was printed on Prusa i3 MK3S 3D-Printer with Poly-lactic acid (PLA) as the filament, and 0% infill. After the 3D-printed hollow bolus structure was ready (Fig 2(B)), it was filled with the prepared Eurosil-4 Pink liquid and was left until the bolus coagulated. After 24 hours, the outer PLA shell was removed to obtain the patient-specific Eurosil-4 Pink bolus (Fig 2(C)). The bolus was found to have a density of 1.15 $g/cm^3$. Two such boluses were constructed with 0.6 cm and 1.1 cm thickness respectively.

## Evaluation of the customised Eurosil-4 Pink boluses using the chest phantom

A patient-specific chest phantom based on the patient's CT was 3D-printed. It was filled with water and sealed. Treatment plans were generated for the chest phantom with 0.6 cm and 1.1 cm Eurosil-4 Pink boluses, 0.5 cm Paraffin wax bolus, and with no bolus. The treatment parameters were the same as the patient plan (6 MV photons, 2 arc-VMAT plan, same pre-scribed dose). Plans were delivered and a Gafchromic EBT3 film (Ashland ISP, Wayne, NJ, USA) was used for dose measurements in each case. (Fig 3) The ethics approval for this project was granted by the Sir Charles Gairdner and Osborne Park Hospital Group as a Quality Improvement Activity (42709). Informed written consent was provided by the patient.

## Results

### Air gap analysis

The variation of surface dose with the bolus thickness is plotted in Fig 4(A), where the dose has been normalised to 100% at the maximum surface dose (Dmax). It was found that a bolus

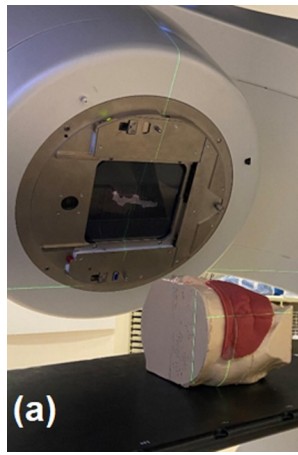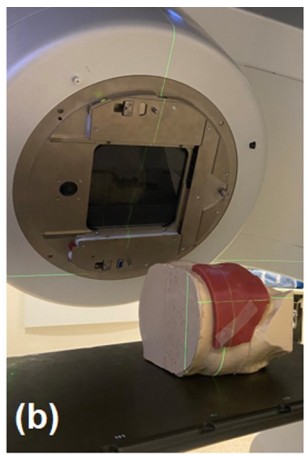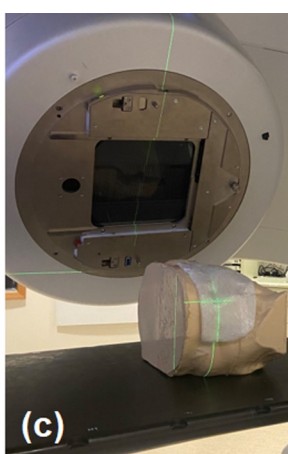

**Fig 3.** Setup to deliver dose to the 3D-printed chest phantom using (a) 0.6 cm customised Eurosil-4 Pink bolus (b) 1.1 cm customised Eurosil-4 Pink bolus (c) Paraffin wax bolus.

thickness of 1.1 cm was most beneficial in providing maximum dose to the surface. This was supported by the TPS predictions and the ion chamber measurements, thereby 1.1 cm was selected as the thickness used for one of the customised Eurosil-4 Pink bolus.

Fig 4(B) shows that the measured surface dose decreased linearly as the thickness of air gap under the bolus was increased in a 6 MV, 10 cm × 10 cm photon beam. The dose is normalised to 100% at the dose measured for no air gap. The dose reductions deduced from Fig 4(B) are summarised in Table 1. It can be inferred that the decrease in dose was not larger than 0.5% even for a 1.5 cm air gap. The results from the TPS were unclear due to the large standard deviation from the mean calculated dose. The average and standard deviation were obtained from contouring the region of the ion chamber during dose calculation.

As displayed in Fig 5(A), there was a considerable reduction in dose for the 2.8 cm × 22.7 cm field compared to the 10 cm × 10 cm field size. Table 1 lists the values for all air gaps. The highest reduction in dose was 2.5% at 1.5 cm air gap for the 2.8 cm × 22.7 cm field. The impact of beam obliquity on dose reduction in the presence of air gaps is demonstrated in Fig 5(B) and Table 2. The values are normalised to 100% at the dose measured for normal incidence. The maximum reduction in dose was observed at 60 degrees incidence angle and was 4.7% when there was no air gap and 6.2% when a 1 cm air gap was present.

### Evaluation of customised Eurosil-4 Pink boluses on the chest phantom

Fig 6 shows that both, the 0.6 cm and 1.1 cm patient-specific Eurosil-4 Pink bolus were able to deliver the prescribed dose (27 Gy) to the target volume by an amount comparable to the Paraffin wax bolus that is commonly used in the clinic. The total delivered dose calculated in the TPS and measured by the radiochromic film is outlined in Table 2 for the four boluses considered in the study. Fig 7 shows the skin as well as the body dose-volume histogram (DVH) for the four cases, and the total prescribed dose has been marked for reference (dashed line). The TPS average dose and standard deviations were obtained from 10 points selected in the film region.

## Discussion

### Air gap analysis

The impact of air gaps under the bolus is not clinically significant when the field is normally incident, and the field size is large. This was shown in the present work by using a 10 cm × 10

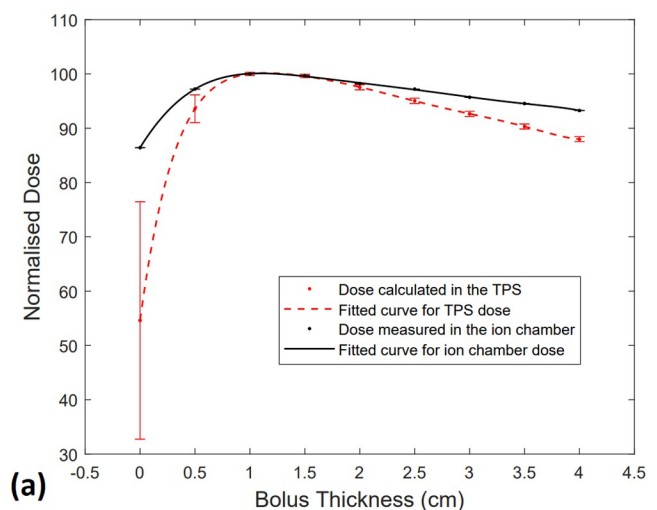
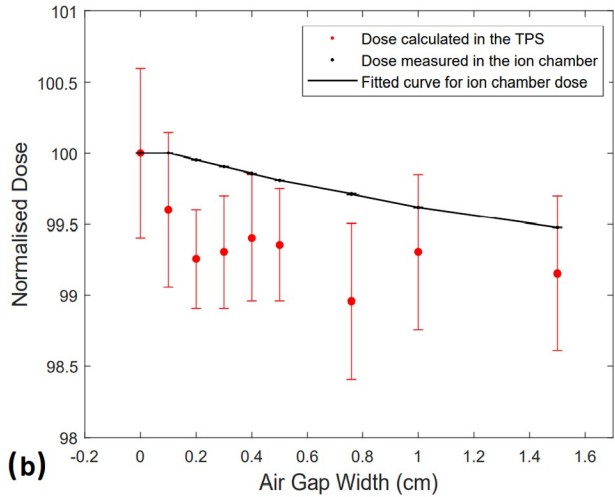

**Fig 4.** (a) Calculated and measured surface dose for bolus thicknesses of 0–4 cm (b) Calculated and measured surface dose for air gap thicknesses of 0–1.5 cm.

cm field size, for which the surface dose did not reduce below 99.5% of the maximum dose, for air gaps between 0.1–1.5 cm. This is consistent with the findings of Khan *et al.* [6]. They described the electron contamination caused by the collimators and solid water in the case of large fields, which dominates the effect of air gaps on the surface dose. In the present study, an effective beam size of 2.8 cm × 22.7 cm was estimated based on the VMAT plan used for the patient. The dose reduction was larger for air gaps above 0.5 cm when this field size was used. It was observed that using different angles of incidence for the effective field size resulted in a significantly larger decrease in surface dose for 1 cm air gap. As shown in Table 3, when the 1 cm air gap was present, the dose was reduced by up to 6.2%, depending on how oblique the beam is. Previous studies have shown a similar trend for obliquely incident beams of small size [4, 10]. Butson *et al.* [4] investigated the effect for an 8 cm × 8 cm field and 1 cm air gap, and reported dose reductions of up to 10%. Chung *et al.* [10] found dose reductions of up to 10.5% for a 6 cm × 6 cm field size and 1 cm air gap. The dose reduction was comparatively less in this study because the longer side of the effective field size could cause larger scatter. However, the overall results indicate that the dose reduction caused by the air gaps tends to be clinically significant for VMAT radiotherapy, and its minimisation needs to be ensured. A further examination using Monte Carlo simulation is possible for this analysis.

**Table 1. Dose reduction with air gap for 10 cm × 10 cm and 2.8 cm × 22.7 cm field size.**

| Air gap thickness (cm) | Reduction in surface dose (%) | |
| --- | --- | --- |
| | 10 cm × 10 cm field size | 2.8 cm × 22.7 cm field size |
| 0.0 | 0.00 | 0.00 |
| 0.1 | 0.00 | 0.00 |
| 0.2 | 0.05 | 0.20 |
| 0.3 | 0.10 | 0.23 |
| 0.4 | 0.14 | 0.33 |
| 0.5 | 0.19 | 0.56 |
| 0.8 | 0.29 | 0.96 |
| 1.0 | 0.38 | 1.29 |
| 1.5 | 0.52 | 2.52 |

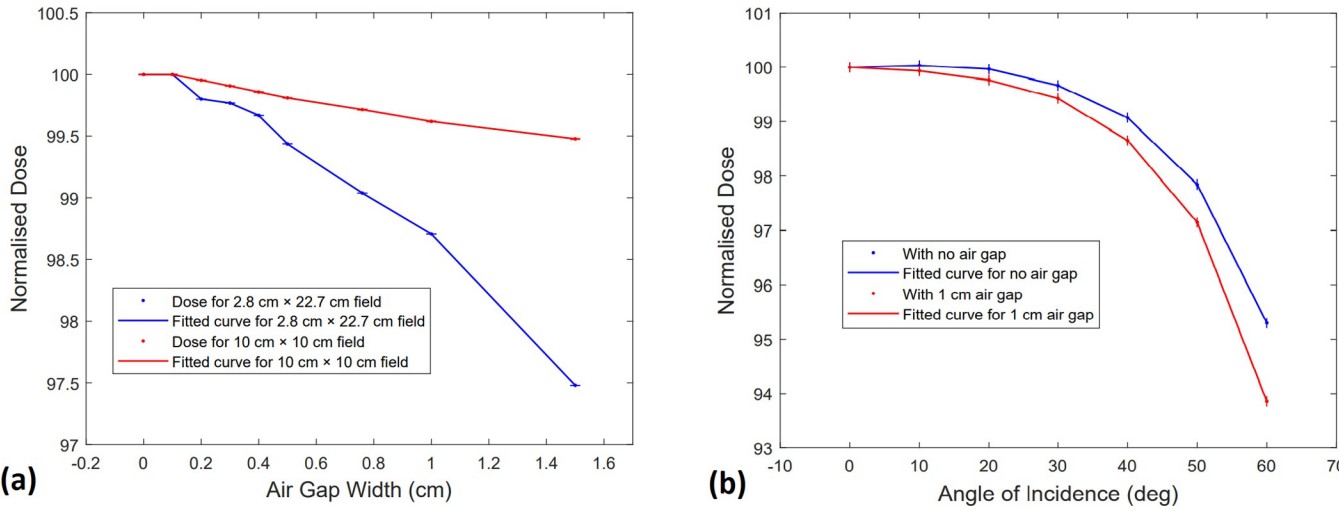

**Fig 5.** (a) Measured dose for air gap thicknesses of 0–1.5 cm for 10 cm × 10 cm (red) and 2.8 cm × 22.7 cm (blue) field sizes (b) Measured dose with and without a 1 cm air gap for angles of incidence between 0–60 degree.

It was observed that the dose calculated near the interface of the air gap and the sheet containing the ion chamber had an unusually large standard deviation in the TPS. The discrepancy between the calculated and measured dose at the air-bolus interface has been investigated by Rana *et al.* [11] for Acuros XB algorithm. It was proposed that this is due to the second build-up region caused by the reduction in scattered radiation in the air gap. Acuros XB may be inefficient at calculating the amount of scattered radiation that reaches the point of measurement lying in the second build-up. The discrepancy is larger for small fields [11].

### Evaluation of the customised Eurosil-4 Pink boluses using chest phantom

The use of 3D-printing in the customisation of bolus can improve the fit to the irregular skin surface, and lower the size and frequency of air gaps [12, 13]; thus, the dose can be delivered more accurately. Kim *et al.* introduced a 3D-printed bolus composed of ABS, and showed that it can shift the maximum dose to the surface, similar to the super flab bolus [7]. It was shown on the RANDO phantom that 3D-printed bolus could be customised to the curved surfaces and act as a suitable build up material. Fujimoto *et al.* provided further evidence for the use of 3D-printed ABS bolus by comparing the PDD curves and DVH parameters of the 3D-printed bolus with the virtual bolus and the commercial bolus from the plans generated for the water phantom and the head phantom [8]. It was shown that the 3D-printed ABS bolus had a PDD curve similar to that of the virtual bolus and the commercial bolus, along with an equivalent dose coverage.

Polylactic acid (PLA) and ABS are the most commonly used materials for making customised boluses due to their water-equivalent densities (1.2 g/cm$^3$ for PLA, 1.04 g/cm$^3$ for ABS)

**Table 2. Calculated and measured dose to the tumour for 0.6 cm and 1.1 cm customised Eurosil-4 Pink boluses, 0.5 cm Paraffin wax bolus and no bolus.**

| Type of bolus used | Calculated dose(Gy) | Measured dose(Gy) | Relative Difference (%) |
|---|---|---|---|
| Eurosil-4 Pink 6 mm | 25.0 ± 1.5 | 28.9 ± 0.2 | 13.5% |
| Eurosil-4 Pink 11 mm | 24.6 ± 1.2 | 29.2 ± 0.3 | 15.8% |
| Paraffin wax | 26.4 ± 0.7 | 29.5 ± 0.3 | 10.5% |
| No bolus | 3.9 ± 6.1 | 18.2 ± 0.2 | 78.6% |

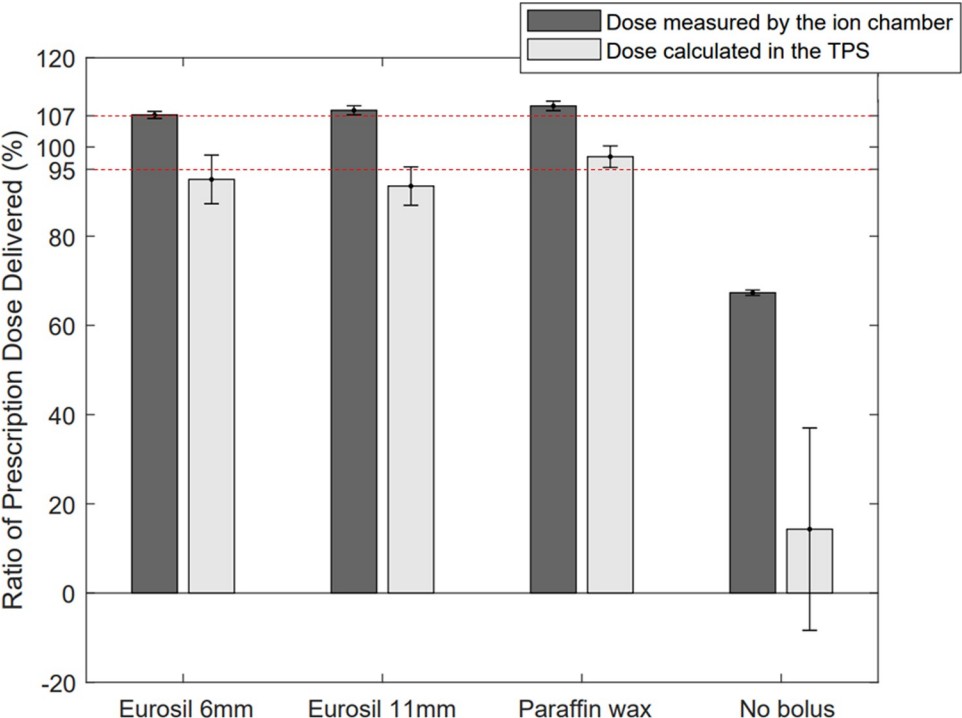

**Fig 6. Comparison of calculated and measured dose for 0.6 cm and 1.1 cm customised Eurosil-4 Pink boluses, 0.5 cm Paraffin wax bolus and no bolus.** Dose limits (95%-107% of prescribed dose) have been marked in red.

and feasibility in 3D-printing. In the present study, Eurosil-4 Pink which is essentially a type of silicone rubber was used to construct the bolus. Silicon rubber has the desired flexibility and surface adhesion needed for conforming to the soft tissue, and it is nontoxic, durable and easy to use [14]. The density of the Eurosil-4 Pink (1.15 g/cm$^3$) is similar to that of PLA, but is a bit higher than ABS. As this is close to the density of water, the bolus satisfies the tissue-equivalence requirement. A previous study by Canters *et al.* highlights the large shift in time from human labour to 3D-printing for a customised Silicone rubber bolus [15]. As per the workflow suggested in the present study, although the total time needed for fabricating the customised Eurosil-4 Pink bolus is 30 hours, only 30 minutes of manual input is necessary since the rest of the process is automated.

The dosimetric evaluation of the customised Eurosil-4 Pink boluses and the Paraffin wax bolus on the chest phantom showed that all the three boluses resulted in nearly the same amount of dose escalation to the tumour (Fig 6). As per ICRU report 50, the dose delivered should not be above 107% of the prescribed dose. It was observed that 0.6 cm Eurosil-4 Pink bolus delivered 5.5 cGy more, 1.1 cm Eurosil-4 Pink bolus delivered 33.5 cGy more and the Paraffin wax bolus delivered 59.5 cGy more than this dose limit as measured by the film. The dose is much higher than the acceptable level, particularly in the case of the Paraffin wax bolus, which can increase the risks of radiation dermatitis and skin cancer. The dose calculated in the TPS was not considered as accurate for this study. The setup for the investigation simulated the real patient case, as the patient CT was used for customisation of the chest phantom and the Eurosil-4 Pink boluses, and the VMAT patient plan was used in the dose delivery. This implies that the customised Eurosil-4 Pink bolus is able to establish sufficient tumour dose coverage to prevent local recurrence, and provide an alternative to the Paraffin wax bolus. For efficient production of the customised bolus, it should be ensured that the hospital staff receive

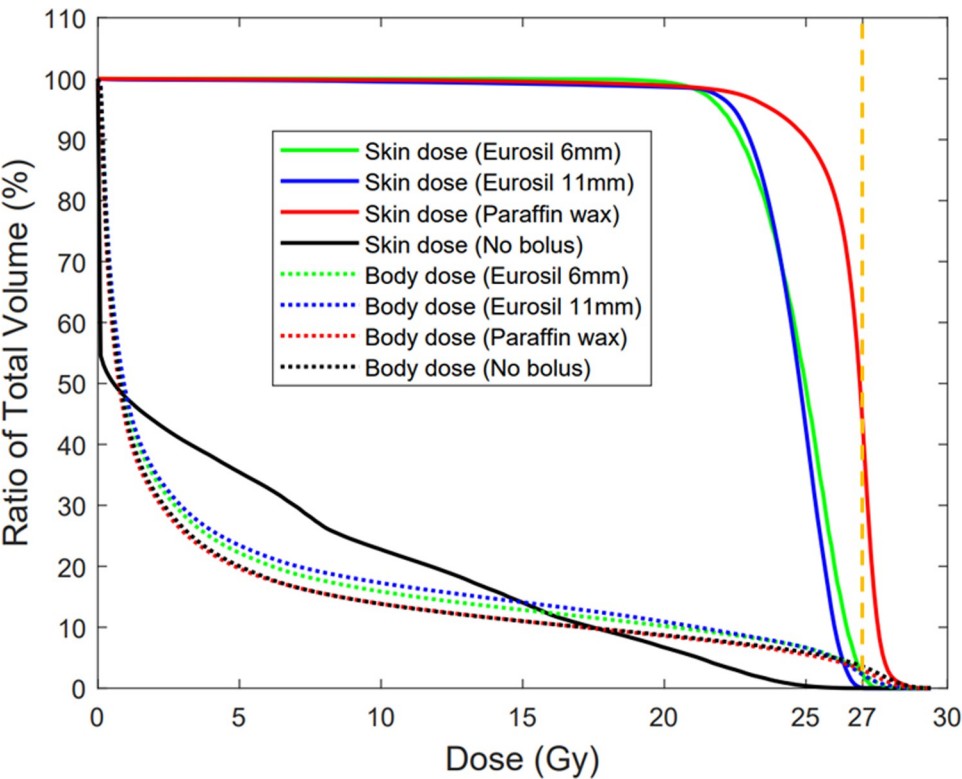

**Fig 7. DVH for skin dose and body dose for 0.6 cm and 1.1 cm customised Eurosil-4 Pink boluses, 0.5 cm Paraffin wax bolus and no bolus.** The prescribed dose (27 Gy) has been marked in orange.

training for 3D-printing. To clinically implement the technique, a quality assurance program can be set up to check the uniformity, density and printing accuracy of the bolus and workflow needs to be documented in the form of an easy to follow work procedure.

## Conclusions

The presence of air gaps between the bolus and the skin may not be a major concern for large fields and normally incident beams. However, in radiotherapy techniques that employ small fields and oblique beams, there is a need for minimising the air gaps in order to ensure sufficient surface dose. Eurosil-4 Pink can be customised to the shape of the patient, and has the desired bolus properties such as malleability and good adhesion to the surface. The customised

**Table 3. Dose reduction with and without 1 cm air gap for 2.8 cm × 22.7 cm field size and angles of incidence between 0–60 degree.**

| Angle of incidence (degrees) | Reduction in surface dose (%) | |
|---|---|---|
| | No air gap | 1 cm air gap |
| 0 | 0.00 | 0.00 |
| 10 | 0.00 | 0.07 |
| 20 | 0.03 | 0.24 |
| 30 | 0.33 | 0.57 |
| 40 | 0.93 | 1.34 |
| 50 | 2.16 | 2.86 |
| 60 | 4.71 | 6.15 |

Eurosil-4 Pink bolus can efficiently shift the maximum dose to the surface, and ensure tumour control. It has the potential to be clinically implemented, and further examination is possible to validate its use.

## Supporting information

**S1 File.**
(PDF)

## Acknowledgments

Special thanks to the department of Radiation Oncology at Sir Charles Gairdner Hospital in Perth for funding this research and providing the equipment. Thanks to Martin Ebert, Talat Mahmood, Mounir Ibrahim, Gabor Neveri, Thomas Milan, Matthew Fernandez and Riley Croxford for the support and productive discussions during the project.

## Author Contributions

**Conceptualization:** Ashlesha Gill, Suki Gill, Pejman Rowshanfarzad.

**Data curation:** Ashlesha Gill.

**Formal analysis:** Ashlesha Gill, Pejman Rowshanfarzad.

**Investigation:** Ashlesha Gill, Warwick Smith, Mahsheed Sabet, Zaid Alkhatib, Suki Gill, Pejman Rowshanfarzad.

**Methodology:** Ashlesha Gill, Andrew Hirst, Mahsheed Sabet, Pejman Rowshanfarzad.

**Project administration:** Pejman Rowshanfarzad.

**Resources:** Andrew Hirst.

**Supervision:** Warwick Smith, Mahsheed Sabet, Zaid Alkhatib, Suki Gill, Pejman Rowshanfarzad.

**Visualization:** Ashlesha Gill.

**Writing – original draft:** Ashlesha Gill.

**Writing – review & editing:** Warwick Smith, Andrew Hirst, Mahsheed Sabet, Zaid Alkhatib, Suki Gill, Pejman Rowshanfarzad.

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
