## [Decision Letter · Decision Letter 0]

2 Feb 2022

PONE-D-21-40497Comparison of conventional versus customized 3D-printed bolus for chest wall radiotherapy in breast cancerPLOS ONE

Dear Ms. Ashlesha Gill,

Thank you for submitting your manuscript to PLOS ONE. After careful consideration, we feel that it has merit but does not fully meet PLOS ONE’s publication criteria as it currently stands. Therefore, we invite you to submit a revised version of the manuscript that addresses the points raised during the review process.

We look forward to receiving your revised manuscript.

Kind regards,

Ngie Min Ung

Academic Editor

PLOS ONE

Journal Requirements:

3. Please amend the Methods section of your manuscript to state either A) that informed consent was provided by the patients, and the type of consent obtained (verbal, written), or B) that the Ethics Committee waived the requirement for informed consent as part of their approval for this study.

Reviewers' comments:

Reviewer's Responses to Questions

**Comments to the Author**

1. Is the manuscript technically sound, and do the data support the conclusions?

Reviewer #1: Yes

Reviewer #2: Yes

2. Has the statistical analysis been performed appropriately and rigorously? 

Reviewer #1: Yes

Reviewer #2: Yes

3. Have the authors made all data underlying the findings in their manuscript fully available?

Reviewer #1: No

Reviewer #2: Yes

4. Is the manuscript presented in an intelligible fashion and written in standard English?

Reviewer #1: Yes

Reviewer #2: Yes

5. Review Comments to the Author

Reviewer #1: Thank you for inviting me to review this paper.

The authors reported the comparison between the standard bolus to the 3D-printed bolus in term of the dose received by the surface and the tumour volume. Overall, this is an excellent paper. It will be great if the authors can provide a clearer pathway for clinical use.

Here are my specific comments:

1. Abstract - “The decrease in surface” decrease for what? Bolus or 3D printed?

2. Abstract - “The dose measured” where?

3. Abstract - “higher for the effective field size and larger angles of incidence” – include some numerical results.

4. Figs and table – can you please limit the number of tables/figures to only 6. As of now there are too many.

5. Please note and comment on the time needed to develop this 3D-printed bolus. This may be an issue given the simplicity of using standard bolus.

6. Please suggest ways to optimise the construction of the bolus so that readers can see clearly on the transition to the clinical use.

7. Last paragraph discussion - Please suggest further studies you recommend before this can be implemented clinically.

Reviewer #2: General comment:

Three-dimensional (3D) printing technology has been applied in many fields including radiotherapy to achieve the goals of optimal dose to the target while sparing the healthy normal tissues. Bolus have been used for decades in radiotherapy and personalized bolus constructed from 3D printing could overcome the shortcoming of the commercial bolus. This article investigated custom made bolus constructed from Eurosil-4 Pink applied in chest wall radiotherapy of breast cancer. The effects are found to be profound for small field sizes and oblique beams. This information extremely crucial when customized bolus is applied for conformal radiotherapy therefore the publication of this article will be a positive addition to the current knowledge in the clinical application of 3D printed bolus in radiotherapy.

The article is generally well written and easy to understand. The article is suitable for publication after a few minor correction

Specific comments:

Title: The title needs to be specific. For example, it is much better to mention the eurosil-4 pink as the bolus materials.

Introduction: Please highlight the shortcoming of current commercial bolus and what is the advantages as well as disadvantages of 3D printed bolus.

Please mention in the details the parameters and the measurements conducted at the last paragraph of the introduction

Materials and methods:

Please write in detail a section on bolus fabrication. It is the Eurosil-4 Pink 3D printed?

How about the tissue equivalency? Did the author conduct any test?

Results:

Table 3: It is paraffin and gauze much better than custom made bolus?

Discussion:

What is the function of silicon rubber as mentioned in sentence line 275?

6. PLOS authors have the option to publish the peer review history of their article (what does this mean?). If published, this will include your full peer review and any attached files.

Reviewer #1: No

Reviewer #2: No

---

## [Author Response · Author response to Decision Letter 0]

7 Apr 2022

Manuscript number: PONE-D-21-40497

The authors would like to thank the Associate Editor and Reviewers for the time and effort on improving this work. It is appreciated and has helped shape a clearer version which we hope you will also find to be improved. 

Reviewer #1: 

Thank you for inviting me to review this paper.

The authors reported the comparison between the standard bolus to the 3D-printed bolus in term of the dose received by the surface and the tumour volume. Overall, this is an excellent paper. It will be great if the authors can provide a clearer pathway for clinical use.

Here are my specific comments:

1. Abstract - “The decrease in surface” decrease for what? Bolus or 3D printed?

This is referring to the decrease in surface dose for the flat Eurosil-4 Pink bolus sheets. Have clarified in the text.

2. Abstract - “The dose measured” where?

Doses were measured by the Gafchromic film under the bolus sheets. Added to the abstract.

3. Abstract - “higher for the effective field size and larger angles of incidence” – include some numerical results.

Supporting numerical data is included.

4. Figs and table – can you please limit the number of tables/figures to only 6. As of now there are too many.

In an attempt to present concisely, the different graphs have been merged for Figure 4 and Figure 5. The number of figures is 7 and the number of tables is 3. Unfortunately, merging or removal of more data seems to degrade the quality of the presentation of results. 

5. Please note and comment on the time needed to develop this 3D-printed bolus. This may be an issue given the simplicity of using standard bolus.

All the steps from designing the bolus from patient CT to importing the G-Code file in the 3D-printer take approximately 30 minutes. With more expertise in using the software such as 3DSlicer, Meshmixer and PrusaSlicer, these steps can be completed in half the time. 3D-printing of the given bolus shell took 5 hours and 24 minutes. This process is largely automatic and doesn’t need much manual input. Filling the bolus shell with the Eurosil-4 Pink bolus can take 10 minutes, and then the drying of the Eurosil-4 Pink takes 24 hours. The bolus can be extracted from the shell in 5 minutes. So, although the total time needed for developing the bolus is about 30 hours, the time required from the hospital staff would only be 30 minutes as the rest of the process is automatic. The Discussion section is modified to include the above. A study by Canters et al. has been added as a reference. 

6. Please suggest ways to optimise the construction of the bolus so that readers can see clearly on the transition to the clinical use.

“For efficient production of the customised bolus, it should be ensured that the hospital staff receive training for 3D-printing. A quality assurance program can be set up to check the uniformity, density and printing accuracy of the bolus.” – these lines are added to the last paragraph of the Discussion section.

7. Last paragraph discussion - Please suggest further studies you recommend before this can be implemented clinically.

Future work has been added at the end of discussion.

Reviewer #2: General comment:

Three-dimensional (3D) printing technology has been applied in many fields including radiotherapy to achieve the goals of optimal dose to the target while sparing the healthy normal tissues. Bolus have been used for decades in radiotherapy and personalized bolus constructed from 3D printing could overcome the shortcoming of the commercial bolus. This article investigated custom made bolus constructed from Eurosil-4 Pink applied in chest wall radiotherapy of breast cancer. The effects are found to be profound for small field sizes and oblique beams. This information extremely crucial when customized bolus is applied for conformal radiotherapy therefore the publication of this article will be a positive addition to the current knowledge in the clinical application of 3D printed bolus in radiotherapy.

The article is generally well written and easy to understand. The article is suitable for publication after a few minor correction

Specific comments:

Title: The title needs to be specific. For example, it is much better to mention the eurosil-4 pink as the bolus materials.

The title is modified to “Comparison of conventional versus customized Eurosil-4 Pink bolus for radiotherapy of the chest wall”.

Introduction: Please highlight the shortcoming of current commercial bolus and what is the advantages as well as disadvantages of 3D printed bolus.

Introduction is modified to address this.

Please mention in the details the parameters and the measurements conducted at the last paragraph of the introduction

These changes have been added in the last paragraph of the introduction.

Materials and methods:

Please write in detail a section on bolus fabrication. It is the Eurosil-4 Pink 3D printed?

How about the tissue equivalency? Did the author conduct any test?

A separate section is added for bolus fabrication, and it is explained more clearly. 

The 3D-printed bolus structure is a customised hollow shell made of PLA. This is filled with Eurosil-4 Pink liquid and after 24 hours when it hardens, the PLA layer is cut and removed. This gives the Eurosil-4 Pink bolus which is customised to the patient. 

The CT of the bolus was examined for calculating the density. It came out to be 1.15 g/cc for the Eurosil-4 Pink bolus, which is close to the density of water. This is added to the Discussion section. 

Results:

Table 3: It is paraffin and gauze much better than custom made bolus?

As per Table 3, the measured and calculated dose are higher for the Paraffin wax bolus (or Paraffin and gauze) bolus as compared to the customised Eurosil-4 Pink boluses. However, the standard deviations values need to be considered along with the average dose values for a full interpretation. As a result, the dose from the table is quite similar for the three boluses. Figure 7 provides useful information for establishing a comparison between the boluses. It is evident that the Paraffin wax bolus exceeds the acceptable dose levels by a much larger amount (59.5 cGy) than the Eurosil-4 Pink boluses. This can cause skin reactions, and the same was observed when the patient was in treatment. Therefore, using the customised Eurosil-4 Pink bolus can be a better option. 

A line is added in the discussion to highlight the issue:

“The dose is much higher than the acceptable level, particularly in the case of the Paraffin wax bolus, which can increase the risks of radiation dermatitis and skin cancer.”. 

Discussion:

What is the function of silicon rubber as mentioned in sentence line 275?

While using Eurosil-4 Pink, the two components (A and B) are mixed together and left out for 24 hours. The solidified Eurosil-4 Pink is essentially a type of silicone rubber. Park et al. [13] pointed out the benefits of using silicone rubber as a bolus material. The same has been mentioned in line 275 of the paper because these advantages are true for the Eurosil-4 Pink bolus. In order to clarify this, the wording was modified.

---

## [Decision Letter · Decision Letter 1]

14 Apr 2022

Comparison of conventional versus customised Eurosil-4 Pink bolus for radiotherapy of the chest wall

PONE-D-21-40497R1

Dear Dr. Gill,

We’re pleased to inform you that your manuscript has been judged scientifically suitable for publication and will be formally accepted for publication once it meets all outstanding technical requirements.

Kind regards,

Ngie Min Ung

Academic Editor

PLOS ONE

Additional Editor Comments (optional):

Reviewers' comments:

Reviewer's Responses to Questions

**Comments to the Author**

1. If the authors have adequately addressed your comments raised in a previous round of review and you feel that this manuscript is now acceptable for publication, you may indicate that here to bypass the “Comments to the Author” section, enter your conflict of interest statement in the “Confidential to Editor” section, and submit your "Accept" recommendation.

Reviewer #1: All comments have been addressed

Reviewer #2: All comments have been addressed

2. Is the manuscript technically sound, and do the data support the conclusions?

Reviewer #1: Yes

Reviewer #2: Yes

3. Has the statistical analysis been performed appropriately and rigorously? 

Reviewer #1: Yes

Reviewer #2: Yes

4. Have the authors made all data underlying the findings in their manuscript fully available?

Reviewer #1: Yes

Reviewer #2: Yes

5. Is the manuscript presented in an intelligible fashion and written in standard English?

Reviewer #1: Yes

Reviewer #2: Yes

6. Review Comments to the Author

Reviewer #1: The authors have adequately addressed my concerns and suggestions.

xxxxxxxxxxxxxxxxxxxxxxxxxxxxxxxxx

Reviewer #2: The authors have address all the suggestion and comments. The article is now acceptable for publication

7. PLOS authors have the option to publish the peer review history of their article (what does this mean?). If published, this will include your full peer review and any attached files.

Reviewer #1: No

Reviewer #2: No

---

## [Editor Report · Acceptance letter]

26 Apr 2022

PONE-D-21-40497R1 

Comparison of conventional versus customised Eurosil-4 Pink bolus for radiotherapy of the chest wall 

Dear Dr. Gill:

I'm pleased to inform you that your manuscript has been deemed suitable for publication in PLOS ONE. Congratulations! Your manuscript is now with our production department. 

Kind regards, 

on behalf of

Dr. Ngie Min Ung 

Academic Editor

PLOS ONE